# Producing Altered States of Consciousness, Reducing Substance Misuse: A Review of Psychedelic-Assisted Psychotherapy, Transcendental Meditation and Hypnotherapy

Agnieszka D. Sekula [1,2,*], Prashanth Puspanathan [2], Luke Downey [1] and Paul Liknaitzky [3,4]

1 Centre for Human Psychopharmacology, Swinburne University of Technology, Hawthorn 3122, Australia
2 Enosis Therapeutics, 2 Gwynne Street, Melbourne 3121, Australia; prash@enosistherapeutics.com
3 Department of Psychiatry, School of Clinical Sciences, Monash University, Melbourne 3168, Australia
4 Turner Institute, School of Psychological Sciences, Monash University, Melbourne 3168, Australia
* Correspondence: agnieszka@enosistherapeutics.com

**Abstract:** A set of interventions that can produce altered states of consciousness (ASC) have shown utility in the treatment of substance misuse. In this review, we examine addiction-related outcomes associated with three common interventions that produce ASCs: psychedelic-assisted psychotherapy (PP), Transcendental Meditation (TM) and hypnotherapy (HT). While procedurally distinct, all three interventions are associated with some common phenomenological, psychological, and neurobiological features, indicating some possible convergent mechanisms of action. Along with addiction and mental health outcomes, these common features are reviewed, and their impact on substance misuse is discussed. While our review highlights some mixed findings and methodological issues, results indicate that PP and TM are associated with significant improvements in substance misuse, alongside improvements in emotional, cognitive and social functioning, behavior-change motivation, sense of self-identity, and meaning. In contrast, and despite its broader acceptance, HT has been associated with mixed and minimal results with respect to substance misuse treatment. Authors identify key research gaps in the role of ASC interventions in addiction and outline a set of promising future research directions.

**Keywords:** substance misuse; addiction; psychedelics; transcendental meditation; hypnosis; hypnotherapy; review; altered states of consciousness; psilocybin; LSD; ibogaine; ketamine; meditation; trance

## 1. Introduction

Addiction is defined as a chronic, compulsive need to engage in a behavior despite its harmful effects and/or the individual's wish to stop [1]. The prevailing mechanistic understanding is of a complex interplay of biological, psychological, and social factors, and available treatments target one or a combination of these determinants. A number of treatment modalities that entail the production of an Altered State of Consciousness (ASC) have shown promise in treating various substance use disorders [2,3], yet the nature and strength of the evidence remain unclear. This review explores treatment outcomes associated with substance misuse interventions that produce ASCs and describes alterations in key biopsychosocial measures relevant to addiction.

While definitions of ASCs vary and are frequently imprecise, we can arrive at a useful operationalization through "change in the overall pattern of subjective experience" [4], a "qualitative... not just quantitative shift" [5], and a "sufficient deviation" [6] within a wide range of mental functions [4], and crucially within "Primary Consciousness" [7]. That is, ASCs are a noticeable (often dramatic) and qualitative alteration to the fundamental "fabric of awareness", typically accompanied by alterations to perception, cognition, and affect. The intentional induction of ASCs through a range of methods is commonplace throughout

history and across cultures. A number of phenomenological and neurobiological features appear common across diverse induction methods such as hypnosis, sensory deprivation, trance, meditation, and psychedelic drugs [8]. These include fragmentation and sometimes loss of a sense of selfhood, changes in the experience of space and time, novel perspectives and the perception of novelty, cognitive changes, perceptual distortion, and emotional lability [9–11].

A wide range of ASC induction methods have been used to treat different forms of substance use disorders within ceremonial, self-medicating, and clinical settings [8]. This review explores three ASC therapies that have been empirically explored as treatments for substance misuse to a greater degree than other interventions: psychedelic-assisted psychotherapy (PP), Transcendental Meditation (TM), and hypnotherapy (HT).

PP involves administration of a psychedelic substance (here, we use the term broadly to include "classical" psychedelics as well as related substances that have substantial overlapping features, including LSD, Psilocybin, Ketamine, Ibogaine, Mescaline, Ayahuasca, and MDMA), usually accompanied by psychotherapeutic or ceremonial support. Within modern clinical trials, psychedelic interventions are typically embedded within a so-called "set and setting" protocol of extra-pharmacological support across at least three distinct treatment phases: preparation, dosing, and integration [12]. An ASC that results from therapeutic doses of a classical psychedelic is frequently reported as one of the most personally meaningful and challenging experiences of an individual's life [13].

Various forms of meditation can be broadly defined as practices of paying attention to present moment percepts in a sustained way and without judgement [14,15]. The TM approach is based on a silent repetition of a personalized mantra [16] and is taught by certified teachers during a standardized four-day induction process. Participants are then expected to continue their practice independently for 15 to 20 min, twice a day. TM emphasizes attainment of "transcendental consciousness", an ASC that is devoid of thoughts or emotions.

Hypnosis is a state of trance, induced by narrowing of attention to specific stimuli with the use of repetitive sounds, mantras or visuals, followed by suggestion of sleep-like relaxation [17]. In a therapeutic setting, the state of relaxation and surrender is leveraged by the hypnotherapist who may persuade immediate or future actions, thoughts, and feelings without requiring the participants' conscious control [18].

Of these induction methods, PP is associated with the most reliable and intense ASCs, to the extent that effective placebo blinding is very difficult to achieve [19]. HT and TM are less reliable in producing ASCs, and the literature associated with these procedures rarely provides clear assessment of the production or characteristics of an ASC. In therapies that employ TM or psychedelics, the acute ASC experience is largely left uninterrupted, with the role of the therapist being limited to supporting or facilitating the experience. Talk therapy takes place predominantly either before or after that experience. In contrast, the state of hypnosis is induced for the duration of talk therapy, and the main therapeutic input occurs during that state.

Herein, we review the literature on the efficacy and related biopsychosocial outcomes associated with PP, TM and HT interventions that target substance misuse. We also consider the role of ASC in precipitating these effects, examine the limitations of each method, and suggest future directions for research.

## 2. Method

Authors reviewed the articles on trials of interventions involving PP (18), TM (10) or HT (13) where at least one outcome related to substance misuse was measured (Figure 1). No restrictions were applied for the type of control condition (including no intervention), type of addiction outcome, or publication date. The following inclusion criteria were applied: (i) human participants, (ii) use of primary data, (iii) empirical neurobiological, psychological or behavioral data, and (iv) peer-reviewed status.

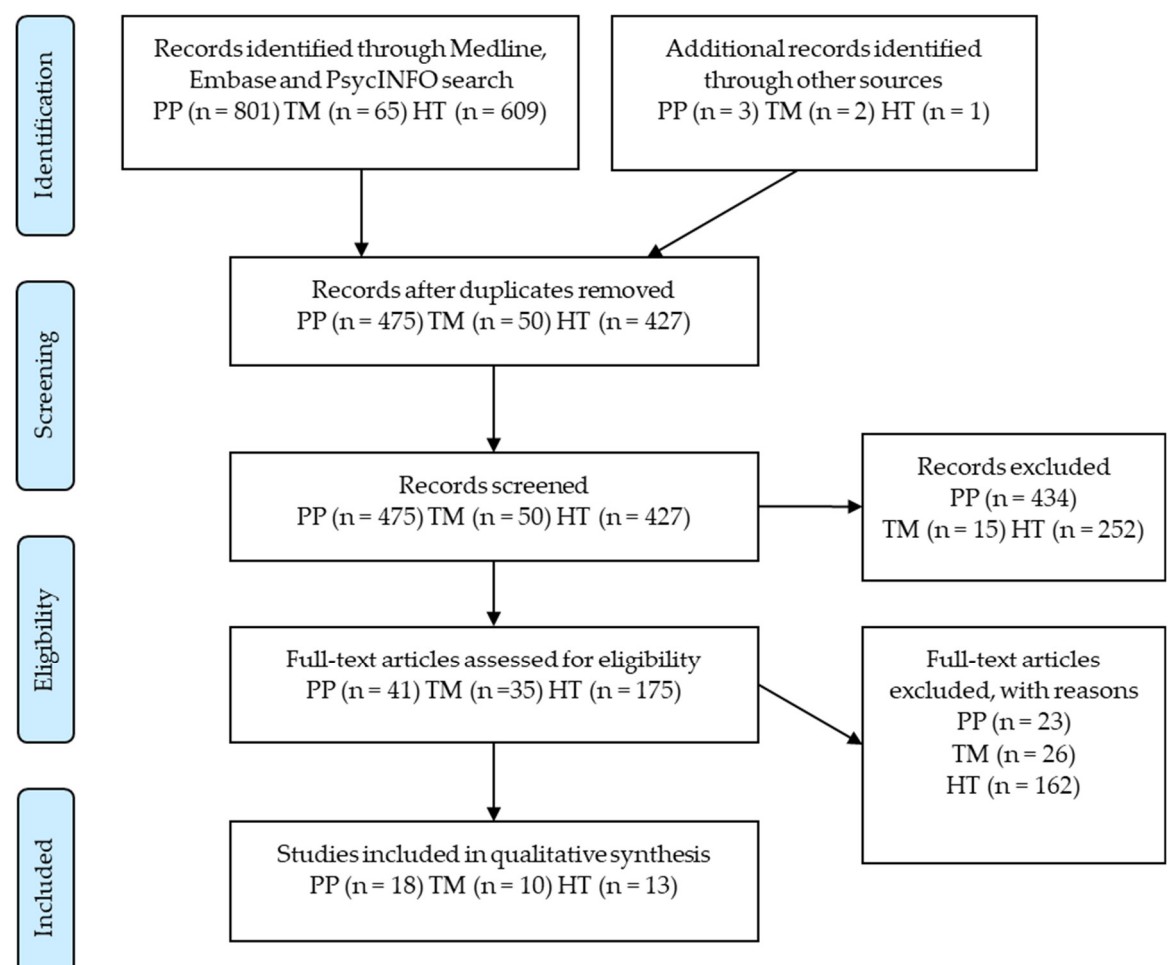

**Figure 1.** PRISMA flow diagram of study selection.

A systematic search was conducted on 10th of August 2020 by two reviewers (ADS, PP) using Medline, Embase and PsycINFO. Abstracts, titles and keywords were searched with the following terms: (addiction OR dependence OR abstinence OR alcohol OR drug OR smoking OR cessation OR cigarettes OR tobacco OR nicotine OR cocaine OR heroin OR methamphetamine OR amphetamine OR stimulant) AND ("Transcendental Meditation" OR Hypnotherapy OR Hypnosis OR LSD OR "Lysergic acid diethylamide" OR Psilocybin OR psilocin OR "Magic mushroom" OR Ketamine OR Ibogaine OR Iboga OR Ayahuasca OR DMT OR dimethyltryptamine OR MDMA OR 3,4-Methylenedioxymethamphetamine OR mescaline OR trimethoxyphenethylamine OR peyote OR "San pedro" OR "2C-B" OR "2,5-dimethoxy-4-bromophenethylamine" OR hallucinogen OR psychedelic).

- Two authors (ADS, PP) scanned titles and abstracts of all articles in search results, independently applying eligibility criteria. Reference lists of included papers were scanned, and an additional six papers were found and manually added. A total of 41 articles that met the inclusion criteria were identified across the three treatment approaches.
- For each included trial, data were extracted independently by the authors. Substance use and abstinence results based on biochemical markers or self-reports were accepted. Quantitative and qualitative reports on biopsychosocial outcomes were extracted from results and discussion sections. Included studies, extracted data and study limitations were compared, and any discrepancies resolved through consensus decision making.

## 3. Results

*3.1. Psychedelic Assisted Psychotherapy*

### 3.1.1. PP Outcomes

The reviewed studies indicated that PP was associated with large, significant reductions in the use of cigarettes [20,21], alcohol [22–24], opiates [25–29], cocaine [30,31], as well as non-significant reductions in cigarette and alcohol use [31]. Significant group effects showed that PP was superior to inpatient psychotherapy [24], outpatient psychotherapy [29], inpatient psychosocial therapy [24] and IV benzodiazepine [23,30].

PP produced significant improvements in key outcome measures directly implicated in abstinence, notably cravings [27,30,32], withdrawal symptoms [25,28,32], and urgency of use [33]. In non-treatment-seeking cocaine addicts, PP induced strong, significant increases in motivation to change [30]. In detoxified, treatment-seeking inpatients, PP aided change by significantly increasing self-reported "self-actualization" (full realization of one's potential) and internal locus-of-control sense of agency over the outcome of life events [26].

Compared to baseline scores, significant improvements were found following PP in comorbid mental health outcomes such as low mood states and depression [27,28,33], state/trait anxiety [27,28,33] and compulsivity [33], alongside qualitative improvements in mood [20,22] and lowering of psychological defense mechanisms [32].

PP was linked to greater insight and internal regulation with significant improvements in understanding of the meaning of life [20,26,27], sense of purpose [26], qualitative improvements in respondents' self-perception [22,31,33], greater insight into one's addiction [32,33], increased urgency for abstinence [33], increased spirituality [20,22], and increased self-efficacy [32]. PP was associated with significant increases in achieving a maximum psychological adjustment score [29], improvements in family/social issues [25], altruistic social effects [22], and interpersonal relationships [20,31].

Krupitsky and colleagues [26,27] investigated the relation between treatment outcomes and dosage/dose frequency, evaluating 2.0 vs. 0.2 mg/kg of ketamine and single vs. three doses repeated at 1-month intervals, and found a significant association between the maintenance of abstinence over 12–24 months and dose strength [26] as well as a higher number of doses [27]. A positive correlation was also observed between intensity of psychedelic experience (strength of peak experiences) and treatment outcomes [34].

Several side effects following the ingestion of a psychedelic compound were reported, and included nausea/vomiting [22,24,33], 20–30% increases in blood pressure and heart rate [20,26,27], anxiety/agitation, which were usually resolved in session with time and interpersonal support [20,23,24], mild headaches post session, which usually resolved within 24 h [20,22], and insomnia on the night post session [22].

One fatality was ruled to be the direct result of ibogaine administration [28]. With no definable pathology identified in postmortem examination, coronial enquiry ruled that this fatality was associated with a failure in the duty of care by the treatment provider. Cardiac arrhythmias related to Ibogaine ingestion have been reported, with pre-existing comorbidities and substance misuse being attributed to the majority of documented deaths [35].

### 3.1.2. PP Limitations

As indicated in Table 1, many of the reviewed trials were open-label with no control groups [25,33], some failed to achieve adequate blinding in placebo-controlled trials [27], some had high rates of drop out 25–50% over 3–18 months [23,24,26], and in some trials the samples were relatively small and homogenous, e.g., in [31]. Due in part to recent media hype and largely positive attitudes towards PP in the general public in places like the US, patients who volunteered for PP may have had strong positive expectancies resulting in a larger placebo component than is typical [20,22].

**Table 1.** Summary of 18 PP studies in addiction that met the inclusion criteria.

| Design | Participants and Tests | Results | Limitations |
|---|---|---|---|
| 1.    **Jensen et al., 1962, USA:** Alcohol Use/Chronic Alcoholics (randomized, longitudinal) [24] | | | |
| **PP**<br>Single-dose 200 mcg LSD + 3 weeks inpatient psychosocial therapy<br>**Control treatment**<br>3-week inpatient psychosocial therapy vs. 3-week inpatient psychotherapy<br>**3 weeks total** | *n* = 125<br>Chronic alcoholic patients of referring counselling centers<br>**Study group: 70**<br>**Control group: 55**<br>**Test** (baseline, 6–18 m)<br>Self-assessment | **Main results**<br>66% patients abstinent in experimental group vs. 41% in psychosocial group vs. 32% in psychotherapy only ($p < 0.05$) | - lack of diagnostic specificity<br>- variable periods of follow-up<br>- large numbers lost to follow-up<br>- lack of clarity on how improvement/abstinence was measured<br>- statistical significance poorly described |
| 2.    **Savage and McCabe, 1973, USA:** Heroin Use/Heroin Dependent Paroled Prisoners [29] | | | |
| **PP**<br>Single session of 300–450 mcg LSD + total of 24 h preparatory psychotherapy over 5 weeks + 1-week inpatient psychotherapy after<br>**Control treatment**<br>4–6-week outpatient psychotherapy<br>**4–6 weeks total** | *n* = 74<br>Heroin-dependent, paroled, male, prisoners matched closely: age, race, religion, marital status, education, years of incarceration<br>**Study Group: 37**<br>**Control group: 37**<br>**Test** (0, 12 m)<br>Psychedelic experience questionnaire; Self-assessment of abstinence + Daily urine test<br>Global adjustment rating scale (via interview with parole officer) | **Main results**<br>Abstinence at 12 m: PAP 25%, control 5% ($p < 0.05$)<br>**Biopsychosocial outcomes**<br>10/10 on global adjustment scale: PAP 32%, control 8% ($p < 0.2$)<br>**Other**<br>12/13 patients with max global adjustment score reported achieving peak experience | - motivations for enrolling the study skewed by likely favorable parole eligibility<br>- sociocultural differences between white therapists and predominantly black inmates/patients<br>- global adjustment rating scale and "psychedelic experience questionnaire" not empirically validated; inconsistent dosing regimens |
| 3.    **Krupitsky et al., 2002, Russia:** Relapse Prevention/Detoxified Heroin Addicts (randomized, double-blind) [26] | | | |
| **High dose PP:**<br>2.0 mg/kg im Ketamine + single-session psychotherapy + existentially oriented psychotherapy pre/post Ketamine + inpatient stay<br>**Active Control:**<br>0.2 mg/kg im Ketamine + single-session Ketamine psychotherapy + existentially oriented psychotherapy pre/post Ketamine + inpatient stay<br>**3–5 days total** | *n* = 70<br>Detoxified, heroin-dependent inpatients at a substance use treatment center, able to provide close affiliate for corroborating data, no significant psychological + craving differences<br>**Study Group: 35**<br>**Control Group: 35**<br>**Test** (Baseline, Day 0, 3 m, 6 m, 9 m, 12 m, 18 m, 24 m)<br>Zung Self-rating Depression Scale (ZDS); Spielberger Self-rating State-Trait Anxiety Scale of Anhedonia Syndrome (SAS); Minnesota Multiphasic Personality Inventory (MMPI); Locus of Control Scale (LCS); Color Test of Attitudes (CTA); Purpose-in-Life Test (PLT); Spirituality Changes Scale (SCS); Urine toxicology | **Main results**<br>Abstinence rates: High-dose PAP 17%, active control 2% ($p < 0.05$) at 24 m<br>**Biopsychosocial outcomes**<br>• Reduction in depression scores at 1 m high-dose PAP > 15% ($p < 0.05$), active control > 20% ($p < 0.001$); at 12 m high-dose PAP > 15% ($p < 0.05$), active control > 29% ($p < 0.05$); no significant changes at 24 m<br>• Significant increases at Day 0 in understanding of the meaning of life ($p < 0.001$); understanding of life purpose ($p < 0.001$); self-actualization ($p < 0.001$); internalization of locus of control ($p < 0.05$). | - no true control group<br>- abstinence verification procedure poorly described<br>- high loss to follow-up by 24 m |

**Table 1.** *Cont.*

| Design | Participants and Tests | Results | Limitations |
|---|---|---|---|
| 4.  **Krupitsky et al., 2007, Russia:** Relapse Prevention/Detoxified Heroin Addicts (randomized) [27] | | | |
| **Single dose PP:** 2.0 mg/kg im Ketamine **Multiple dose PP:** 0 m, 1 m, 2 m: 2.0 mg/kg im Ketamine **Other** 5 h preparatory psychotherapy at 0 m +Pre-dose addiction counselling at 1 m, 2 m +1 h post-dose psychotherapy at 0 m, 1 m, 2 m | *n* = 53 Inpatients at a substance use treatment center, detoxified and abstinent >2 weeks, with heroin dependence for at least 1 year, no significant psychological + craving differences **Single dose PP: 27** **Multiple dose PP: 26** **Test** (Baseline, Day 0, monthly for 12 m) Physical exam; Urine toxicology; ZDS, SAS, PLT and VASC; telephone interview, self-report assessment via Timeline Follow-Back technique | **Main results** Abstinence rates at 12-month follow-up: multiple-dose PP 50% vs. single-dose PP 22% ($p < 0.05$) **Biopsychosocial outcomes** Significant improvements at 12-month f/u ($p < 0.005$) in depression; state and trait anxiety; cravings for heroin; understanding the meaning of life | - no placebo control group - no adequate blinding was feasible |
| 5.  **Thomas et al., 2013, Canada:** Poly-Substance Use/First Nations Community Volunteers (observational) [31] 6.  Follow-up: Argento et al., 2019 [36] | | | |
| **PP** - Ayahuasca brew (non-quantified) on Days 1 + 3 - Residential group treatment + sweat lodge ceremony + unstructured dialogue **Other** Ritualistic setting overseen by indigenous shamans **4 days total** | *n* = 12 Non-abstinent, non-treatment seeking, polysubstance using volunteers **Test** (Baseline, Day 1, Week 2, monthly on Month 1–6) Difficulty in Emotion Regulation Scale (DERS); Philadelphia Mindfulness Scale (PHLMS); Empowerment Scale (ES) Hope Scale (HS); McGill Quality of Life survey (MQL); 4-Week Substance; Use Scale (4WSUS); Semi-structured Qualitative Interview | **Main results** Substance use (average 4WSUS score 6 m f/u vs. baseline): Tobacco 18% reduction, alcohol 30% reduction, cocaine ($p < 0.05$) 60% reduction, hallucinogens 9.1% increase, opiates, cannabis and sedatives nil change. **Biopsychosocial outcomes** - Statistically significant improvements ($p < 0.05$) in mindfulness; empowerment; hopefulness; quality of life—meaning; quality of life—outlook - Improvements in sense of spirituality; connection with self, others and nature | - small sample size - homogenous sample population - no control group - no standardization of dose - ritualistic context may have a confounding effect size - low rigor of statistical analysis - no record of any other concurrent treatments - some participants were repeat attendees |

**Table 1.** *Cont.*

| Design | Participants and Tests | Results | Limitations |
|---|---|---|---|
| 7. **Anja Loizaga-Velder and Rolf Verres 2014, Mexico:** Poly-Substance Use/ Participants of Ayahuasca Ritual (retrospective) [32] | | | |
| **PP** Ayahuasca brew, differing protocols | *n* = 14 Substance-dependent volunteers, mean age 42 years, 2 years post-Ayahuasca ceremony **Test** (>2 years post treatment) Unstructured interview | **Main results (no statistical analysis)** <br>• Reduction in cravings: 64.2% <br>• Attenuation of withdrawal symptoms: 21.4% <br><br>**Biopsychosocial outcomes** Better understanding of the underlying causes of addiction; improvements in self-efficacy; transformation of consciousness to help overcome cravings; lowered psychological defense mechanisms | - small sample size <br>- homogenous sample population <br>- no control group <br>- non-randomized, purposefully chosen sample <br>- no standardization of original intervention <br>- no statistical analysis of results |
| 8. **Dakwar et al., 2014, USA:** Cocaine Cravings/Non-Treatment Seeking Cocaine Addicts (cross-over design, blinded) [30] | | | |
| **PP** Randomized to: <br>• Ketamine, low dose, 0.41 mg/kg <br>• Ketamine, high dose, 0.71 mg/kg <br>• Lorazepam 2 mg as IV infusion <br>• Repeat infusions separated by 48 h of all test substances via crossover design <br>• Inpatient treatment + 10 min mindfulness exercise pre/post dose <br>**Other** Day 1–3: Inpatient achievement of abstinence **9 days total** | *n* = 8 Non-abstinent, non-treatment seeking, cocaine-dependent volunteers **Test** (Baseline, Day 1, weekly on Week 1–4) University of Rhode Island Change Assessment (URICA); Visual Analog Scale of Craving (VASC); psychiatrist interview of abstinence via Timeline Follow-Back technique; urine toxicology | **Main results** <br>• Cue-induced cravings (Day 1 Median VASC score): Low-dose Ketamine −126 vs. Lorazepam 65 ($p < 0.012$). Higher dose augmented these effects, not statistically significant. <br>• Follow-up average drug use: At 28 days USD 10.50/day vs. Baseline USD 149.30/day ($p < 0.001$) <br><br>**Biopsychosocial outcomes** Motivation to change (Day 1 Median URICA score): Low-dose Ketamine 3.6 vs. Lorazepam 0.15 ($p = 0.012$) | - small sample size <br>- homogenous sample population <br>- no placebo control group <br>- short interval between 1st and 2nd dose is likely to confound 2nd dose results |
| 9. **Johnson et al., 2014, USA:** Tobacco Smoking/Nicotine Dependent Volunteers [20] <br>10. Follow-up 1: Garcia-Romeu et al., 2014 [34] <br>11. Follow-up 2: Johnson et al., 2017 [21] <br>12. Follow-up 3: Noorani et al., 2018 [37] | | | |

**Table 1.** *Cont.*

| Design | Participants and Tests | Results | Limitations |
|---|---|---|---|
| **PP** 20–30 mg/70 kg Psilocybin at Week 5, Week 7 and optional at Week 13 + Week 1–4: Weekly CBT for smoking cessation + Week 5–15: Weekly supportive psychotherapy **15 weeks total** | *n* = 15 Nicotine-dependent volunteers, >10 cigarettes per day, multiple unsuccessful quit attempts, seeking to quit **Test** (Weekly on week 0–15, 6 m): Breath Carbon Monoxide level Urine Cotinine level; self-report; States of Consciousness Questionnaire (SOCQ) | **Main results** Abstinence: 80% at 6 m, 67% at 12 m, 60% at >12 months ($p < 0.05$) **Biopsychosocial outcomes** <br>• Abstainers (all *p*'s < 0.05): 13.7% more psilocybin occasioned mystical experience (SOCQ); 17.6% higher personal meaning and 15.6% greater spiritual significance <br>• positive mood changes, improvements in interpersonal relationships, spirituality, and ambivalence to change | • small sample size <br>• no control group <br>• self-selection bias <br>• inconsistent dosing regimens |

13. **Bogenschutz et al., 2015, USA:** Alcohol Use/Alcohol Dependent Volunteers [22]
14. Follow-up: Bogenshutz et al., 2018 [38]

| Design | Participants and Tests | Results | Limitations |
|---|---|---|---|
| **PP** 0.3 mg/kg Psilocybin at 4 weeks + 0.3–0.4 mg/kg at 8 weeks + 7 sessions of motivational interviewing + 3 preparatory sessions + 2 debriefing sessions post PP **12 weeks total** | *n* = 10 Alcohol-dependent, concerned about their drinking, not currently in treatment, abstinent and not in withdrawal **Test** (Baseline, Weeks 0, 4, 5, 8, 9, 12, 24, 36) SCID for DSM IV Addiction Research Center Inventory (ARCI) Alcohol Abstinence Self-Efficacy Scale (AASE) Profile of Mood States (POMS) Short Inventory of Problems (SIP) Timeline Follow-Back (TLFB) procedure self-assessment of drinking Breath Alcohol Concentration (BAC) | **Main results** <br>• Weeks 5–12 compared to baseline: 26% decrease in HDD (Heavy Drinking Days) and 27.2% decrease in DD (Drinking Days) (both $p < 0.01$). <br>• Weeks 5–12 compared to <br>• Weeks 1–4: 18.2% decrease in HDD ($p = 0.026$) and 21.9% decrease in DD ($p < 0.05$). <br>• Week 36 compared to baseline: <br>• >50% decrease in DD and HDD ($p < 0.01$) **Biopsychosocial outcomes** positive mood changes, positive attitudes about the self and life, altruistic social effects, and increased spirituality; maintained at up to 12 months. | - small sample size <br>- no control group <br>- no biological verification of reduced use/abstinence |

**Table 1.** *Cont.*

| Design | Participants and Tests | Results | Limitations |
|---|---|---|---|
| **15.** **Mash et al., 2018, USA:** Heroin and Cocaine Addiction/Patients Seeking Detoxification [33] | | | |
| **PP**<br>Single-dose Ibogaine 10 mg/kg + inpatient stay<br>**12 days total** | *n* = 191<br>102 Opiate-dependent and 89 Cocaine-dependent active users self-referred for detoxification<br>**Test** (Baseline, Days 0, 5, 30)<br>Heroin (HCQ-29) and Cocaine (CCQ-45) Craving Questionnaires<br>Beck Depression Inventory version II (BDI-II); Profile of Moods (POMS, 2nd edition); Symptom Checklist—90 scales (SCL-90) | **Main results**<br>Significant improvements in urgency of use and ability to quit ($p < 0.0001$) at Day 30<br>**Biopsychosocial outcomes**<br>• Significant improvements in compulsivity and negative mood states ($p < 0.0001$) at Day 30<br>• Improvements in renewed sense of self, increased insight, and urgency for abstinence | • no control group<br>• self-selection bias<br>• psychedelic/ psychosocial effects confounded by physiological effect of Ibogaine on withdrawal |
| **16.** **Noller et al., 2018, New Zealand:** Opiate Addiction/Opioid Dependent Volunteers (observational) [28] | | | |
| **PP**<br>Single-dose Ibogaine 25–55 mg/kg + inpatient stay<br>**Other**<br>Inpatient stay:<br>Provider 1: >1 week (*n* = 1)<br>Provider 2: <4 days (*n* = 13)<br>**Up to 1 week total** | *n* = 14<br>Opiate-dependent volunteers seeking treatment, able to provide close affiliate for corroborating data, recruited from 2 different ibogaine treatment providers<br>**Test** (Baseline, Day 0, 3 m, 6 m, 9 m, 12 m)<br>Subjective Opioid Withdrawal Scale (SOWS); Addiction Severity Index Lite (ASI-Lite); Beck Depression Inventory-II (BDIII); urine screens | **Main results**<br>• Withdrawal symptom 44% reduction at Day 0 ($p = 0.004$)<br>• Addiction severity: 80% reduction ($p = 0.004$) at 12 m<br>**Biopsychosocial outcomes**<br>Depression scores: >50% reduction ($p = 0.013$) at 1 m, 80% reduction ($p = 0.004$) at 12 m | - small sample size<br>- selection bias from patient sample actively seeking Ibogaine treatment<br>- inconsistency in matching of substance use reports and biological verifications<br>- post-treatment protocols varied between 2 providers<br>- inconsistent dosing regimens |
| **17.** **Brown and Alper 2018, USA:** Opiate Addiction/Patients Seeking Detoxification (observational) [25] | | | |
| **PP**<br>Single dose of 1540 mg Ibogaine + inpatient treatment<br>**Other**<br>Stabilized for 3 days pre-dose on short acting opioid<br>**3–6 days total** | *n* = 30<br>Heavy and relatively selective users of opioids, actively seeking treatment, able to provide close affiliate for corroborating data, recruited from 2 different ibogaine treatment providers<br>**Test** (Baseline, Day 0, 3 m, 6 m, 9 m, 12 m)<br>Addiction Severity Index Lite (ASI-Lite); Subjective Opioid Withdrawal Scale (SOWS) | **Main results**<br>• Withdrawal symptoms > 50% reduction ($p < 0.001$) at Day 0<br>• Addiction severity: >50% reduction ($p < 0.001$) at 12 m<br>**Biopsychosocial outcomes**<br>Family/social issues: >80% reduction ($p < 0.001$) at 12 m | - small sample size<br>- no control group<br>- no biological verification of self-report metrics<br>- recent long-acting opioid use confounding consistency of withdrawal profile |

**Table 1.** *Cont.*

| Design | Participants and Tests | Results | Limitations |
|---|---|---|---|
| 18. **Dakwar et al., 2020, USA:** Alcohol Use/Treatment Seeking Alcoholics (randomized, blinded) [23] | | | |
| **PP** Ketamine 0.71 mg/kg as IV infusion **Control** Midazolam 0.025 mg/kg as IV infusion **Other** 6-session motivational interviewing **5 weeks total** | *n* = 40 Non-abstinent, treatment seeking, alcohol-dependent volunteers **Study Group: 17** **Control Group: 23** **Test** (Baseline, Day 1, weekly on Week 1–5, 6 months) Clinical Institute Withdrawal Assessment; Alcohol Abstinence Self-Efficacy Scale; Perceived Stress Scale; Five Facet Mindfulness Questionnaire; Barrett Impulsiveness Scale; psychiatrist interview of abstinence via Timeline Follow-Back technique; urine toxicology | **Main results** <br><br> • Abstinence (6 m, *n* = 19, *p* < 0.05): Ketamine 75%, Midazolam 27% <br> • Alcohol use (over 21 days post infusion, *n* = 34): Ketamine 47,1%; Midazolam: 59.1% <br> • Presence of a heavy drinking day (over 21 days post infusion, *n* = 34): Ketamine 17.6%; Midazolam: 40.9% <br> • Probability of a heavy drinking day with each post-infusion day (*p* < 0.05): Ketamine: OR = 0.98, 95% CI = 0.89–1.08, *p* = 0.74; Midazolam: OR = 1.19, 95% CI = 1.14–1.25, *p* < 0.001 <br> • Time to relapse (log rank test, *p* = 0.04): Ketamine group significantly longer than Midazolam <br><br> **Biopsychosocial outcomes** <br> No significant differences in craving, withdrawal, mindfulness, impulsivity, stress sensitivity, and self-efficacy | - small sample size <br> - homogenous sample population <br> - no placebo control group <br> - high rates of dropout at 6 m f/u <br> - 6 m f/u did not follow TLFB technique <br> - results of psychological measures not clearly reported |

Several studies were inconsistent in their dosing regimen or lacked details regarding dosage and timing of drug delivery [20,28,29,31,32], lacked clarity on how outcome measures were obtained or verified [23,24,26], or did not report concurrent psychotherapy or other treatments [31,32]. Early studies and ritualistic settings used vague diagnostic metrics [24,29,31,32]. Many recent studies predominantly used self-report measures without biological verification [22,24–26,31–33]. In one study, biological verification was inconsistent with self-reports [28].

There were also notable differences in treatment approach across reviewed studies, with clinical vs. ritualistic settings [31,32] and varying forms and amounts of psychotherapeutic support [20,22,24,26,27,29]. Such heterogeneity limits the ability to draw overall conclusions about therapeutically important aspects of the treatment approach.

*3.2. Transcendental Meditation*

3.2.1. TM Outcomes

As indicated in Table 2, TM led to significant reductions from baseline in the use of cigarettes [39,40], alcohol [39,41–43], prescribed psychotropics [39,44], and most illicit drugs [39,45] except for hallucinogens [39]. Significant group effects showed that TM

was superior to no treatment [39,40], counselling [41,43,45], exposure therapy [44], cranial electrical stimulation [43], and psychiatric medication [42].

**Table 2.** Summary of 10 TM studies in addiction that met the inclusion criteria.

| Treatment | Participants and Tests | Results | Limitations |
|---|---|---|---|
| 1    **Ballou, 1977, USA:** Drug Dependance/Prisoners (randomized, longitudinal) [46] | | | |
| **TM**<br>4-day instruction + group f/u daily/6 weeks; weekly/3 m; bimonthly until 10 m + individual sessions available + drug dependence treatment<br>**Control**<br>drug-dependence treatment (unspecified)<br>**10 months total** | *n* = 66<br>drug-dependent inmates, meditation-naive; stopped intake of illegal drugs ≥ 15 days prior<br>**Study group: 30** interested in meditation<br>**Control group 1: 20** interested in meditation<br>**Control group 2: 16** uninterested in meditation<br>**Test** (baseline, 10 m) Spielberger State/Trait Anxiety Inventory | **Main results**<br>TM: 2 reduced all substances, 6 ceased all drugs; 6 reduced smoking, 2 ceased smoking<br>**Biopsychosocial outcomes**<br><br>• Anxiety ($p < 0.001$): trait: TM −21%, controls—no change; state: TM −17%, control—1 + 14%, control 2—no change<br>• Activities: TM increase from 2.88 h/week at baseline to 10.75 h/week at 10 m<br>• 23 of TMs presented qualitative reports: more calm, relaxed, peaceful, outgoing (23); improved relationships, increased understanding, compassion, social interest, decreased outbursts of temper, social apprehension (20); feeling better physically (18)<br><br>**Other**<br><br>• Rule infractions: TM baseline 0.15/m, 10 m 0.036/m; controls—no change;<br>• TM practice: 13 regular, 5 irregular, 5 occasional; out of 23 qualitative reports: desire to continue TM after release and support implementing it nationwide (23); believe TM enabled psychological (17), physical (16), substance abuse (14) improvements; satisfied—18, not satisfied—2 with TM | - no control group data on substance use or activities<br>- substance abuse: unreliable in prison due to fear of repercussions<br>- no baseline data on substance use<br>- TM is an adjunct to a drug-dependence treatment |

**Table 2.** *Cont.*

| Treatment | Participants and Tests | Results | Limitations |
|---|---|---|---|
| 2 **Brautigam, 1977, Sweden:** Drug Abuse/Youth (randomized, blinded) [45] | | | |
| **TM** 4-day instruction + group f/u 2 h weekly/1 m + counselling: 4 h biweekly **Control** 3-month counselling **Other** Therapist in both groups: psychiatrist or psychologist **6 months total** | *n* = 20 Youth **Study group: 10** **Control group: 10** Each group: 6 hashish only and 4 hard drug users (LSD, opiates, amphetamines); 5 drug-related convictions **Test** (0, 3, 6 m) behavioral observation Leisure time: 0—no time, 3—much time | **Main results** (all *p*'s < 0.05)<br>• Hashish, person/month: TM at baseline 19.2, at 3 m—3, at 6 m—regulars 3 (stable), non-regulars 11 (increasing); controls at baseline—20.5; at 3 m—18.2<br>• Hard drugs, person/month: TM at baseline—2.4; at 3 m—0.2; at 6 m—regulars 0.6, non-regulars 1.4 (increasing); controls at baseline—3; at 3 m—6.8<br>• Biopsychosocial outcomes<br>• Leisure time: TM +0.55; controls: +0.14<br>• TM vs. control (*p* < 0.05): increased stability, adjustment skills (self-acceptance, satisfaction, copying), confidence; decreased tension, restlessness, psychomotor retardation, marked decrease in anxiety (controls showed the opposite trend)<br>• TM reported being more energetic, active, selective in choice of TV/books/staying friends with drug users; increased joy and positive contacts with others<br>**Other** f/u attendance: all TMs, 7/10 controls | - small, unrepresentative sample size<br>- vague outcome measures |

**Table 2.** *Cont.*

| Treatment | Participants and Tests | Results | Limitations |
|---|---|---|---|
| 3 **Monahan et al., 1977, USA:** Substance Use Prevention/General Population (retrospective) [39] | | | |
| **TM** Personal TM instruction, applied at various times by various teachers **Control** no treatment | $n = 417$ members of Philadelphia World Plan Centre of the International Meditation Society **Study group: 264** (194 active meditators/70 no longer meditating) **Control group: 153** non-meditating friends of study group members **Test** (3 m before treatment, post treatment) Average weekly substance use | **Main results** Before treatment/post treatment <br>• nicotine: all TMs: 2.1/1.7 ($p < 0.05$); active TMs: 2.2/1 ($p < 0.05$); control: 1.71/1.75 ($p > 0.05$) <br>• soft alcohol: all TMs: 2.5/2.1 ($p > 0.05$); active TMs: 2.5/1.7 ($p < 0.05$); control: 2.9/3 ($p > 0.05$) <br>• hard alcohol: all TMs: 2.7/1.7 ($p < 0.05$); active TMs: 3/1.8 ($p < 0.05$); control: 3.25/3.2 ($p > 0.05$) <br>• marijuana: all TMs: 3.6/1.5 ($p < 0.05$); active TMs: 3.3/0.9 ($p < 0.05$); control: 4/3.3 ($p > 0.05$) <br>• hallucinogens: all TMs: 0.2/0.05 ($p > 0.05$); active TMs: 0.2/0.03 ($p > 0.05$); control: 0.2/0.1 ($p > 0.05$) <br>• stimulants: all TMs: 1/0.2 ($p > 0.05$); active TMs: 0.95/0.05 ($p < 0.05$); control: 0.5/0.2 ($p > 0.05$) <br>• sedatives: all TMs: 0.3/0.04 ($p > 0.05$); active TMs: 0.25/0.005 ($p > 0.05$); control: 0.8/0.3 ($p > 0.05$) <br>• opiates: all TMs: 0.4/0.02 ($p < 0.05$); active TMs: 0.05/0.005 ($p > 0.05$); control: 0.1/0.04 ($p > 0.05$) <br>• prescription drugs: all TMs: 2.6/1.7 ($p < 0.05$); active TMs: 3/1.6 ($p < 0.05$); control: 1.2/1.6 ($p > 0.05$) | - no description of intervention procedures - low return rate of questionnaires (22.3%) - retrospective study, limited variable control |

**Table 2.** *Cont.*

| Treatment | Participants and Tests | Results | Limitations |
|---|---|---|---|
| **4** | **Brooks and Scarano, 1985, USA:** Alcohol Use/PTSD Patients (randomized, blinded) [41] | | |
| **TM** 4-day instruction, 1.5 h/day + group f/u 1 h/weekly for 3 m Qualified instructor, staff instructed on method **Control** individual psychotherapy + group counselling weekly/ 3 m **3 months total** | *n* = 18 male war veterans, all motivated, blind to treatment type **Study group: 10** **Control group: 8** **Test** (baseline, 3 m) PTSD Figley Scale, Taylor Manifest Anxiety Scale, Beck Depression Inventory, Figley post-Vietnam Adjustment scale (1—most severe, 4—no problem), stress copying: audio stimulus habituation via skin resistance (stimulus GSR) | • Main results<br>• TM vs. control ($p < 0.05$): reduced alcohol consumption F (1,19) = 16.5<br>• Figley adjustment scale (alcohol dependence): TM at baseline—2.0, at 3 m—3.7; controls unchanged<br>• Biopsychosocial outcomes<br>• post-Vietnam adjustment scale for TMs (all $p$'s < 0.05): insomnia at baseline—2.7, at 3 m—3.7; family problems at baseline—2.1; at 3 m—3.3; at 3 m, depression scale −54%; anxiety scale −45%;<br>• PTSD symptom scale (DSM III) −30%; emotional numbness scale −54%; controls unchanged (group effect $p < 0.05$)<br>• Habituation/stress copying ($p > 0.05$): TM −44%, controls +20%<br>• 7/10 TMs reported improving enough to no longer need medical services | - small, homogenous sample size<br>- no info on previous treatment |

**Table 2.** *Cont.*

| Treatment | Participants and Tests | Results | Limitations |
|---|---|---|---|
| 5     **Tuab et al., 1994, USA:** Alcohol Use Disorder Relapse Prevention/Inpatients (randomized) [43] | | | |
| **TM** <br> Routine treatment + TM: 3 prep meetings + 1 individual session + 3 group sessions + 2 × 20 min TM meditation for 20 days; f/u 1/month Certified instructors <br> **Control 1** <br> Electromyographic biofeedback, 1 h/day + 20 min/day self-practice for 20 days <br> **Control 2** <br> Neurotherapy, 30 min for 15 days <br> **Control 3** <br> 3-month routine treatment: AA and alcoholism counselling <br> **18 months total** | *n* = 118 <br> inpatients, long history of abuse; 1-week detoxification <br> **Study group: 35 (f/u 32)** <br> **Control group 1: 24 (f/u 22)** <br> **Control group 2: 28 (f/u 26)** <br> **Control group 3: 31 (f/u 25)** <br> **Test** (baseline; 1–6 m; 7–12 m; 13–18 m) <br> Social history Questionnaire, Beta Intelligence Tests, Profile of Mood States | **Main results** (group effects at all time points $p < 0.05$) <br> % of days at baseline/1–6 m/ 7–12 m/13–18 m: <br> • not drinking: TM 26.2/71.7/64.6/76.3; Control 1 21.3/68.0/66.8/79.2; Control 2 28.1/59.8/55.3/60.5; Control 3 31.8/50.0/52.9/45.7 <br> • 1–2 drinks: TM 6.7/0.4/0.8/0.5; Control 1 7.2/0.5/0.7/0.5; Control 2 3.6/9.3/3.6/3.3; Control 3 3.4/11.3/3.8/4.4 <br> • 3–6 drinks: TM 11.3/3.3/5.0/3.4; Control 1 12.1/3.7/4.7/2.9: Control 2 8.0/9.8/13.2/11.8; Control 3 7.5/11.9/13.9/16.1 <br> • 6+ drinks: TM 55.8/24.6/29.6/19.8; Control 1 59.4/27.8/27.8/17.4; Control 2 60.3/21.1/27.9/24.4; Control 3 57.3/26.3/29.4/33.8 <br><br> **Biopsychosocial outcomes** <br> TM improved from baseline ($p < 0.05$) on tension–anxiety, depression–dejection, anger–hostility, vigor activity, fatigue–inertia, confusion–bewilderment. <br> **Other** <br> Adherence: TM—90.2%; Control 1—88.6%; Control 2—94.8% | - biopsychosocial measure report vague |
| 6     **Royer, 1994, USA:** Smoking Cessation/General Population (longitudinal,) [40] | | | |
| **TM** <br> 4-day instruction 1.5 h/day, voluntary f/u <br> **Controls** <br> None <br> **4 days total** | *n* = 324 <br> volunteers; 20% tried professional methods to quit <br> **Study group:** 110 <br> **Control group:** 214 <br> no significant differences on demographic, smoking measures, motivation, attempts to quit <br> **Test** (baseline, 20–24 m) smoking and adherence | • Main results <br> • Group effects for quit + reduced smoking $p < 0.05$ <br> • Quit smoking: fully adherent TM—55%, partially adherent TM—21%, controls—21% <br> • Reduced smoking: fully adherent TM—26%, partially adherent TM—34%, controls—12% <br> • Other <br> • TM adherence: 33% completely (20 min/2 × day), 67% partially <br> • Compliance rate: no sig diff between smoking meditators and non-smoking meditators (288) | - mailed-out, self-report questionnaires <br> - non-randomized <br> - no report on the relationship between the desire to quit and outcome <br> - no information on TM instruction procedure or instructor |

**Table 2.** *Cont.*

| Treatment | Participants and Tests | Results | Limitations |
|---|---|---|---|
| 7   **Haaga et al., 2011, USA:** Substance Use/Youth (randomized, blinded) [47]<br>8   **Nidich et al., 2009, USA:** Substance Use/Youth (randomized, blinded) [48] | | | |
| **TM**<br>90 min intro + 10 min interview + 1 personal + 3 group instructions; f/u: individual 30 min/week in 1st month then once/month; voluntary weekly group meetings; teachers certified by Maharishi Mahesh Yogi (1970s), 6 m training, >35 y experience, recertified in 2005<br>**Control**<br>none<br>**Other**<br>participants and assessors blind to study aim and conditions<br>**3 months total** | *n* = 207<br>students, all substances used; non-sig diff on demographics and substance use; matched on ADHD and gender<br>**Study group: 93**<br>**Control group: 114**<br>**Test** (baseline, 3 m)<br>substance use inventory, Profile of Mood States (total mood disturbance scale + tension/anxiety, depression/rejection, anger/hostility) constructive Thinking Inventory | **Main results**<br>• no group effects ($p > 0.05$) for cigarette, drug or alcohol abstinence and cigarette or drug use<br>• sig group effects ($p < 0.05$) on alcohol use among men: TM at baseline—7.07, at 3 m—5.83; controls at baseline—8.67/week, at 3 m—10.11/week<br>• Biopsychosocial outcomes<br>• TM vs. control ($p$'s $< 0.01$): improved distress, anxiety, depression, anger/hostility, coping ability<br>• Among hypertension individuals TM vs. control ($p < 0.05$): TM −5.0, controls +1.3 mmHg DBP; TM −2.8, controls +1.2 mmHg CBP<br><br>**Other**<br>TM adherence (once/day) 65% | - smoking and drug use already low at baseline<br>- smoking and drug use banned in restricted university environment, drinking age 21<br>- motivation to decrease substance use not measured/unlikely<br>- very low adherence<br>- possible instructor effect<br>- measurements of preceding week only, likely obscured information (e.g., weekend binge) |
| 9   **Barnes et al., 2016, USA:** Psychotropic Medications Dependance/PTSD Patients (retrospective) [44] | | | |
| **TM**<br>Prolonged exposure OR cognitive processing therapy + TM: 5 days, 6 × 2 h individual sessions + group meetings + voluntary f/u (1st month weekly, 2–6 m monthly)<br>Certified teachers, Maharishi Foundation<br>**Controls**<br>Prolonged exposure OR cognitive processing therapy<br>**6 months total** | *n* = 74<br>active-duty military service members with PTSD or Anxiety Disorder; inpatients, completed traumatic brain injury therapy<br>**Study group: 37**<br>**Control group: 37**<br>matched on age, sex, diagnosis and baseline medication use<br>**Test** (baseline, 1, 2, 3, 6 m)<br>Medication: prescription refill; total mg/week; changes in symptoms (distress, interpersonal functioning, social role) from baseline (<1 decrease, >1 increase) | **Main results**<br>Psychotropic medication use:<br>• stabilized/decreased/ceased: at 1 m, TM—83.7%, controls—59.4% ($p < 0.05$); at 2 m, no sig diff; at 3 m, TM—75.6%, controls—59.4% ($p < 0.05$); at 6 m, no sig diff<br>• increased: at 1 m, TM—10.8%, controls—40.5% ($p < 0.05$); at 2 m, no sig diff; at 3 m, TM—5.4%, controls—40.5% ($p < 0.05$); at 6 m, no sig diff<br>• new meds introduced: at 1 m, TM—5.4%, controls—32.4% ($p < 0.05$); at 2 m, no sig diff; at 3 m, TM—2.7%, controls—27.0% ($p < 0.05$); at 6 m, no sig diff<br><br>**Biopsychosocial outcomes**<br>(all $p$'s $< 0.05$)<br>Psychological symptom severity: at 1 m, TM—0.86, controls—1.25 ($p < 0.05$); at 2 m, no sig diff; at 3 m, TM—0.9, controls—1.05 ($p < 0.05$); at 6 m, no sig diff | - participation based on completion of TM training and self-report of regular TM practice (once per day, 5 days per week) for at least 3 months following the start of training<br>- motivation bias<br>- symptom severity measures vague<br>- single value for all types/strengths of meds<br>- retrospective study, limited variable control<br>- possible non-compliance/misuse<br>- to match controls, charts were reviewed over long timespan, possible treatment changes |

**Table 2.** *Cont.*

| Treatment | Participants and Tests | Results | Limitations |
|---|---|---|---|
| **10**　**Gryczynski et al., 2018, USA:** Alcohol Use Disorder Relapse Prevention/Inpatients [42] | | | |
| **TM**<br>Residential treatment + TM: 4 days of 1 h intro; 1 individual + 3 group sessions; voluntary f/u weekly/12 weeks at local centers or over the phone Certified instructors<br>**Control**<br>3–4-week integrated substance use disorder residential treatment: medically managed withdrawal, structured activities, group and individual counselling, cognitive–behavioral counselling, 12-step approach, relapse prevention<br>**3 months total** | *n* = 50<br>AUD-diagnosed inpatients, newly admitted, those that completed medically assisted withdrawal, meditation-naive, prisoners and those with severe mental conditions excluded; non-sig diff on psychological + craving measures; TM cohort vs control: drinking (20.2 vs. 25) and heavy drinking (18.6 vs. 24) days/m at baseline<br>**Study group: 26**<br>**Control group: 24**<br>**Test** (baseline, 3 m)<br>Timeline follow-back questionnaires; alcohol consumption; Addiction Severity index Lite, Perceived Stress Scale, Alcohol Urge and Craving Experience; helpfulness of the TM scale: 0—not at all, 10—extremely | **Main results**<br><br>• at 1 m ($p > 0.05$): alcohol use: TM—35%, control—38%; heavy drinking: TM—19%, control—25%<br>• at 3 m ($p > 0.05$): alcohol use: TM—42%, control—54%; heavy drinking: TM—6%, control—10%<br>• at 3 m ($p < 0.05$): alcohol use: regular TMs—25%, others—59%; heavy drinking, regular TMs—0%, others—47%,<br>• biopsychosocial outcomes<br>• non-sig group diff ($p > 0.05$): perceived stress −38%, psychological distress −36%, alcohol urge −35%, craving strength −42%, craving frequency −61%;<br>• qualitative reports (10 is max): helpfulness of TM in reducing stress: 8.4 (35% rated 10); dealing with alcohol cravings: 7.8 (35% rated 10); preventing or limiting alcohol use: 7.8 (38% rated 10)<br><br>**Other**<br><br>• outpatient f/u attendance similar (50–60%); TM adherence: 38% 2 × day, 62% 1 × day for >15 days<br>• controls: 46% meditated despite being instructed not to | - robust nature of facility program undermines between-group effects<br>- inpatient study—difficult to generalize<br>- non-randomized sample<br>- TM adherence vs. outcomes unavailable (greater adherence to recommended TM practice was significantly correlated with better outcomes across a range of measures) |

Only two studies showed non-significant group effects. Among students who perceived no negative consequences of substance use, TM was associated with reductions only in alcohol use and only among men [47]. The use of other substances and substance-by-gender interactions were not significantly different between TM and no treatment groups. Nonetheless, everyone in the TM group improved significantly on biopsychosocial measures, including anxiety, depression, hostility, and coping, while controls worsened [48]. In another study, a robust residential program combined with pharmacological management of withdrawal symptoms was equally effective to TM in alleviating psychological symptoms of AUD, including alcohol urge, craving strength, and craving frequency [42]. A notable limitation was that half of the controls in this study practiced meditation despite being instructed not to, potentially compromising the comparison condition.

Numerous other biopsychosocial measures improved following TM, while controls showed no change [43,46] or more frequently worsened [41,43–46,48]. TM led to significant improvements in functions critical to addressing addiction, most prominently coping skills [45,47,48]. Alcohol use, relapse rates, and withdrawal symptoms improved among AUD patients [42,43]. TM practice was an effective replacement for pharmacological treatment of PTSD, leading to reduced use of prescription medication and alcohol along

with significant reductions in PTSD symptoms; the comparison condition—exposure with cognitive processing therapy—led to increased use of medication and worsening of psychological symptom severity [44]. Compared with counselling, TM was associated with greater improvements in stress habituation, PTSD symptoms including emotional numbness, and adjustment scores, including insomnia, employment status and family problems [41]. TM also led to improvements in general psychological well-being, most frequently significant and substantial reductions in stress and anxiety, e.g., in [41,45], depression, e.g., in [41,43,48] and anger, e.g., in [43,45,46]. Significant improvements were also observed in interpersonal relationships, including increased compassion and social interest as well as reduced hostility [41,43,46–48]. Numerous participants reported feeling better physically, with reductions in insomnia, fatigue, psychomotor retardation, and hypertension [41,43,45–48]. Adherence to TM was also linked to increased participation in other beneficial activities, including sport and reading [43,46].

Higher frequency of TM practice contributed to better substance use outcomes [39,40,42,45], but not biopsychosocial outcomes, which improved even following irregular practice, e.g., in [42]. Participants generally attributed the major source of their improvements to TM, e.g., in [42,46], thought it should be implemented at other prisons nationwide [46] and often returned to practice after relapsing [45]. Others requested to start the TM program sooner than anticipated [45] or extend the intervention phase [46].

Except for studies that recruited participants after they already attended a TM program [39,42,44], motivation was controlled for through randomization [41,43,45–48] or matching controls for motivation [40]. When reported, adherence rates were comparable between control conditions and TM practitioners, e.g., in [40,42].

### 3.2.2. TM Limitations

Potential conflicts of interest were apparent across a number of trials. Two studies were co-funded by a TM-promoting foundation [42,44] and in several studies at least one author was affiliated with a private TM institution [39,42,43,47,48], although these affiliations do not necessarily imply biased results.

Several studies used TM as an adjunct to some other treatment, making it difficult to distinguish TM's unique impact [42,43,45,46]. While TM induction is standardized and the 4-day instruction program is comparable across studies, control conditions were not always clearly described, especially in older studies, e.g., "counselling" [39–41,45,46].

Most researchers relied solely on self-reports (although validated questionnaires were usually employed), without physiological markers or behavioral observations [39,40,42,43, 46,47]. Moreover, addiction endpoints were heterogeneous, limiting comparisons across studies.

### 3.3. Hypnotherapy
### 3.3.1. HT Outcomes

As summarized in Table 3, all reviewed HT studies led to small [49–53] or moderate [54–59] improvements in substance use outcomes from baseline, although significance levels were not provided.

**Table 3.** Summary of 13 HT studies in addiction that met the inclusion criteria.

| Treatment | Participants and Tests | Results | Limitations |
|---|---|---|---|
| **1 Schubert, 1983, USA:** Smoking Cessation/General Public (randomized) [59] | | | |
| **Hypnotherapy** (4 × 50 min, weekly) Suggestions on misconceptions about the self in relation to smoking **Control** (4 × 50 min, weekly) (1) Systematic relaxation + suggestions on misconceptions about the self in relation to smoking (2) no treatment **4 weeks total** | *n* = 70 >3 years of smoking, currently > 15 cigs/day, no predominant mental illness or current psychotherapy **Study group: 22** **Control group 1: 19** **Control group 2: 29** **Test** (baseline, 4 m f/u) Smoking Cessation QA; Harvard Scale of Hypnotic Susceptibility | **Main results** <ul><li>Abstinence at 4 weeks ($p > 0.05$): 55% hypnotherapy, 74% controls (1); ($p < 0.05$) 0% controls (2)</li><li>Abstinence at 4 m f/u ($p > 0.05$): 55% hypnotherapy, 58% controls (1); ($p < 0.05$) 7% controls (2)</li></ul> **Other** Smoking reduction at 4 weeks and 4 m ($p < 0.05$): Ps in upper 2/3 on hypnotic susceptibility rating—14.5% greater in hypnotherapy vs. controls (1) | - no information on nature of hypnotic suggestion - no details on hypnotic induction procedure - no information about hypnotherapist - clinical measures used not reported - minimal statistical comparison with passive control group |
| **2 Manganiello, 1984, USA:** Methadone Addicts/Inpatients (randomized) [58] | | | |
| **Hypnotherapy** (30–60 min, 2 × week) psychotherapy + HT standard trance induction + hypnotic suggestion (desensitization of drug cue) + self-hypnosis training **Control** (30–60 min, 2 × week), psychotherapy, individual sessions **6 m total** | *n* = 69 Inpatients; post 6 m methadone treatment, detoxification, no psychosis/impending incarceration **Study group: 35** **Control group: 34** **Test** (baseline, 1, 6, 9, 12 m) Symptomatic complaints (severity on scale 1–3); urinary analysis; methadone med logs | **Main results** (all *p*'s < 0.05) <ul><li>Achieving withdrawal: 45.7% hypnotherapy, 0% controls</li><li>methadone dose level: lower for hypnotherapy vs. controls (at 6 m and f/u 6 m post treatment)</li><li>abstinence from illicit use: 57% hypnotherapy, 20% controls</li></ul> **Biopsychosocial outcomes** ($p < 0.05$) discomfort and withdrawal symptoms stronger for controls: trouble sleeping, no appetite, nervousness, anxiety, body aches | - no details about therapist - hypnotherapy not a standalone treatment - study selected only those subjects who demonstrated stability and abstinence from illicit drug use before induction into the study - no details on hypnotic induction procedure |
| **3 Hyman et al., 1986, Australia:** Smoking Cessation (randomized) [57] | | | |
| **Hypnotherapy** (4 × 60 min) induction to trance state; suggestions to emphasize negative aspects of smoking **Control** (4 × 60 min) (1) Focused cessation: rapid cessation technique (3 × 15 min/session) (2) Attention placebo: discussion of topics of concern to the subject (3) No treatment **4 weeks total** | *n* = 60 avg 30 cigs/day, non-sig diff on demographics and smoking rates **Study group: 15** **Control group 1: 15** **Control group 2: 15** **Control group 3: 15** **Test** (post treatment; 3 and 6 m f/u) Smoking QA; serum thiocyanate level | **Main results** (all *p*'s > 0.05) <ul><li>Cigarettes/day post treatment/ 3 m/6 m: hypnotherapy—4.8/15.2/14.1; control (1)—8.3/14.5/15.6; control (2)—6.1/13.5/11.1</li><li>Abstinence post treatment/ 3 m/6 m: hypnotherapy—60%/40%/40%; control (1)—50%/35%/20%; control (2)—50%/40%/20%</li></ul> | - baseline smoking questionnaire not described - no information on hypnotherapist - assessment of abstinence rates poorly described - limited results of controls |

**Table 3.** *Cont.*

| Treatment | Participants and Tests | Results | Limitations |
|---|---|---|---|
| **4** **Sorensen et al., 1995, USA:** Smoking Cessation/Employees [53] | | | |
| **Hypnosis** (90 min) 3 hypnotic exercises + behavioral strategies + videotape for self-practice at home; group-based **Other** Smoking ban was introduced at the workplace **Single treatment** | $n$ = **4367** (f/u 2642) Employees, 17% previously attended structured cessation program **Test** (baseline, 16 m) Cessation survey | **Main results** Abstinence: 15% **Other** 71% of all smoking employees attended at least 1 session; 80% reported quitting because of smoking ban | - no control condition - hypnosis combined with introduction of smoking ban at a workplace policy - no details on the hypnosis session - no info on the hypnotherapist |
| **5** **Ahijevych et al., 2000, USA:** Smoking Cessation/General Public [49] | | | |
| **Hypnosis** (60 min) hypnosis included relaxation (deep breathing, concentration on self-efficacy phrases and being in control of situations, 40 min) + audiotape (progressive muscle relaxation, breathing, self-hypnosis induction, repetitions of positive attitude phrases, 9 min) + info on smoking risks (20 min) hypnotherapist: clinician, >15 years of hypnosis experience **Single treatment** | $n$ = **452** volunteers attending American Lung Association hypnosis session; 79% attempted to quit previously **Test** (5–15 m post instruction) over-the-phone smoking QA | **Main results** • 22% quit in a month prior to the survey; 65% had a smoke-free period since instruction • Av. nr of cigarettes/day: before HT—27 (SD = 13.5), after HT—21 (SD = 12.7) **Other** • Impact of perceived ease of hypnotizability: ×2 $(4, n = 432) = 23.56, p < 0.05$. • Participants reported relaxation as the most helpful aspect of the program | - no control condition - 17% had previous experience with hypnotherapy; 55% rated themselves as easily hypnotized - 59% simultaneously used other strategies (including 26% nicotine replacement therapy and 6.2% oral substitutes) - paid participation (USD 40) |
| **6** **Pekala et al., 2004, USA:** Alcohol Use Disorder/Inpatients [60] | | | |
| **Hypnotherapy** (4 × 50 min) intensive therapy + self-hypnosis training audiotape records + slow deep breathing with hypnotic suggestions: ego strengthening, relapse prevention, serenity enhancement, anger and anxiety reduction **Control** (4 × 50 min) (1) intensive therapy + stress management program; (2) intensive therapy + transtheoretical cognitive–behavioral program; (3) intensive therapy only intensive therapy: group + individual; 5 days/week, 6 h/day, **21–28 days total** | $n$ = **261** (f/u 141) Inpatients of substance abuse rehabilitation program **Study group: 41** **Control group 1 + 2: 36** **Control group 3: 64** **Test** (baseline; 2 m) Addiction Severity Index (ASI); State Self-Esteem Scale (SSES); Relapse Prevention Assessment Inventory (RAPI); hypnotizability (PCI-HAP); States of Change Readiness and Treatment Eagerness Scale (SOCRATES); hypnotic susceptibility (predicted Harvard Group Scale, pHGS) | **Main results** Abstinence ($p > 0.05$): 87% total abstinence, 9% relapse to drugs or alcohol, 4% relapse to both **Biopsychosocial outcomes** (all $p$'s < 0.05) • Self-esteem: regulars (3–5/weeks) > controls; • Anger/impulsivity: controls > regulars, irregulars > controls **Other** • Practicing self-hypnosis predicted by pHGS (coeff. 0.28), ideomotor finger item (coeff. −0.30), use of counterconditioning (coeff. 0.41), and recognition of problem (coeff. −0.32); R = 0.703 • Abstinence predicted by self-esteem (coeff. −0.81), practice of self-hypnosis tapes (coeff. 0.36), serenity (coeff. 0.50), counterconditioning (coeff. 0.51), and stimulus control (coeff. −0.28); R = 0.793 | - 13% dropped out, further 46% lost to follow-up - results on other conditions are not reported - interventions were not compared (only total data for abstinence reported; different psychological measures used) - hypnotherapy not a standalone treatment - highly homogenous sample (gender) - no information about hypnotherapist |

**Table 3.** *Cont.*

| Treatment | Participants and Tests | Results | Limitations |
|---|---|---|---|
| 7     **Elkins et al., 2006, USA:** Smoking Cessation/General Public (randomized) [54] | | | |
| **Hypnotherapy** (8 × 90 min) Counselling (30 min) + mental imagery (60 min) + 4 × 5–10 min supportive phone calls Eye-focus induction + cessation suggestions + posthypnotic visualization of cessation benefits Hypnotherapist: clinical psychologist or physician, 40 h HT training provided by the PI **Control** Self-help materials + 4 × 5–10 min supportive phone calls **8 weeks total** | *n* = 20 >10 cigs/day, keen to quit, no other substance abuse, no NR **Study group: 10** **Control group: 10** **Test** (pre/post treatment, weeks 12, 26 f/u) Fagerström Test for Nicotine Dependence (FTND); expired carbon monoxide (CO); self-report of last 7 days smoking | **Main results** <br>• Abstinence at 12 weeks f/u ($p < 0.05$): 60% hypnotherapy, 0% controls; at 26 weeks f/u ($p < 0.05$): 40% hypnotherapy, 0% controls <br>• Average number of cigarettes per day at 26 weeks f/u: 3 hypnotherapy, 20 controls | - hypnotherapy not a standalone treatment - highly homogenous sample (gender, race, education) - small sample |
| 8     **Barabasz et al., 1986, USA:** Smoking Cessation/General Public [50] | | | |
| **Hypnotherapy** (60–90 min) Intro+ imagination exercise + individualized induction based on eyeball set induction (30 ss) + progressive relaxation + depth suggestions (1–4 min) + instructions for smoking cessation (4 min) + self-hypnosis intro Groups 1 and 2—individual; Group 3—group-based; Group 4—individual + 1–3 90 min f/up sessions; Group 5—individual + 1–3 90 min restricted environmental stimulation therapy; Group 6—individual, without individualized induction process Hypnotherapist: Groups 1, 3, 4, 5—experienced clinician; Groups 2, 6—intern **Control** Intro session (experienced clinician) **1–3 sessions** | *n* = 307 volunteers **Study group 1: 83** **Study group 2: 45** **Study group 3: 66** **Study group 4: 20** **Study group 5: 30** **Study group 6: 16** **Control group: 47** **Test** (6–30 m post instruction) Tellegen Absorption Scale; Beck Depression Scale; Stanford Hypnotic Clinical Scale (SHCS) | **Main results** Abstinence: Group 1: 28%; Group 2: 13%; Group 3: 36%; Group 4: 30%; Group 5: 47%; Group 6: 4%; control: 6%. **Other** <br>• Abstinence correlated with lower Beck depression scores at baseline ($p < 0.05$), higher Tellegen Absorption Scale scores ($p < 0.05$) and higher SHCS scores ($p < 0.05$) <br>• Treatment satisfaction scores correlated with nr of months of abstinence ($p < 0.05$) | - very small control group - paid participation (USD 76 to USD 296) - voluntary group selection |

**Table 3.** *Cont.*

| Treatment | Participants and Tests | Results | Limitations |
|---|---|---|---|
| 9     **Hely et al., 2011, Australia:** Smoking Cessation/General Public [52] | | | |
| **Hypnosis** Phenomenology of Consciousness Inventory—Hypnotic Assessment Procedure (PCI-HAP) + self-hypnosis instructions: audio record of cessation info + CBT exercises + hypnotic suggestion (relaxation, visualization, and anchoring instructions) + intervention diary **Single treatment** | *n* = **11 (f/u = 7)** volunteers **Test** (baseline, 6 weeks) Fagerstrom Test for Nicotine Dependence; PCI-HAP assessment; alveolar carbon monoxide (CO) levels; hypnotizability, hypnotic expectancy; MSPSS (social support); DASS: Depression, Anxiety and Stress | **Main results** 1 abstinence, 2 reduced smoking, 4 unchanged **Other** <br>• No correlation between cigarette use results and nicotine dependence, hypnotizability, social support, depression, stress, or anxiety scores. <br>• Hypnotizability scores mild to moderate | - no control condition - psychological measures provided only at baseline - no statistical analysis - no control group - 4/11 did not complete the study |
| 10     **Golabadi et al., 2012, Iran:** Opioid Addiction/Outpatient (randomized) [55] | | | |
| **Hypnotherapy** (5 × 45 min) Psychotherapy + HT + self-hypnosis (15 min/day) Progressive relaxation, eye fixation, deep breathing, counting down, imagery visualization + therapeutic suggestions (visualizing successful opium cessation, control over oneself, mental and physical health, opium's harmful effects, dislike and nausea when smelling opium) Hypnotherapist: clinician experienced in addiction cessation **Control** (5 × 45 min) Psychotherapy, consultation with clinical psychologist **1 m total** | *n* = **21** completed detoxification, non-sig diff between groups (baseline characteristics), no drugs other than opium for 1 m prior to admission, completed detox for opium, no history of mental retardation/active psychosis **Study group: 10** **Control group: 11** **Test** (baseline, 6 m) Urine test; general behavioral questions (asked participant + reliable person, e.g., family member) | **Main results** relapse rate ($p > 0.05$): 40% (4/10) hypnotherapy; 73% (8/11) controls **Biopsychosocial outcomes** withdrawal symptoms (nr of patients reporting symptoms before/after hypnotherapy): Restlessness: 10/8; Insomnia: 8/7; Bodily pain: 5/3; Autonomic disturbance: 6/2 | - no information about hypnotherapist - no group comparison on subjective reports - incomprehensive tests: no scale (yes/no questions on withdrawal symptoms) - hypnotherapy not a standalone treatment - small, homogenous sample (gender) |

**Table 3.** *Cont.*

| Treatment | Participants and Tests | Results | Limitations |
|---|---|---|---|
| 11 **Dickson-Spillmann et al., 2013:** Switzerland: Smoking Cessation (randomized) [51] | | | |
| **Hypnotherapy** (100 min) Psycho-Education (40 min) + guided imagery HT (40 min) with suggestion to disconnect pleasant experiences from smoking, self-image of non-smoker, dealing with temptation and withdrawal symptoms, evoke positive commitment to cessation, assume responsibility + debriefing (20 min) + CD for home use Hypnotherapist: male hypnosis and relaxation therapist **Control** (100 min) Psychoeducation (40 min) + relaxation (40 min) with suggestion (the same) + debriefing (20 min) All group-based **Single session** | *n* = 223 (f/u 186) >5 cigs/day, wanting to quit, not using NR or other substances, no psychotic symptoms **Study group: 116 (f/u 99) Control group: 107 (f/u 87) Test** (baseline, 2 weeks & 6 m f/u) Fagerström Test for Nicotine Dependence (FTND); Beck Depression Inventory-V (BDI-V); Beck Anxiety Inventory (BAI); 12-Item Short Form Health Survey (SF-12); Salivary cotinine levels; smoking abstinence and self-efficacy; Minnesota Nicotine Withdrawal scale (MNWS) | **Main results** • Abstinence at 2 weeks ($p > 0.05$): 33.3% hypnotherapy, 24.5% controls; at 6 m ($p > 0.05$): 14.7% hypnotherapy, 17.8% controls. • Mean nr of cigarettes at 2 weeks ($p > 0.05$): 9.4 hypnotherapy, 8.9 controls; at 6 m ($p > 0.05$): 13.6 hypnotherapy, 14.3 controls. • MNWS scores ($p < 0.05$): 0.6 hypnotherapy, 0.63 controls **Biopsychosocial outcomes** No intervention effect or group differences on self-efficacy or adverse effects | - little information about hypnotherapist - hypnotherapy not a standalone treatment - anxiety and depression scores on f/u measured but not reported - de-blinding post intervention led to disappointment in controls, potentially lowering motivation |
| 12 **Hasan et al., 2014, USA:** Smoking Cessation/Inpatient (randomized) [56] | | | |
| **Hypnotherapy** (90 min) Trance (repetitive statements and deep breathing) + relaxation + phone counselling (5 × 15 min) + tape for self-hypnosis + (1) behavioral counselling (30 min) and nicotine replacement NR (1 m) OR (2) behavioral counselling (30 min) Suggestions: visual imagery on health, building self-worth and urge control, negative affectivity toward nicotine, dissociating pleasant experiences from smoking Hypnotherapist: certified hypnotist and a tobacco treatment specialist **Control** (30 min) (1) behavioral counselling (30 min) + NR (1 m) + self-help materials; (2) no treatment **Single session** | *n* = 155 (f/u 99) Inpatients with a cardiac or pulmonary illness, exclusion: terminal illness, other substance abuse, psychiatric illness, HT or NR within last 5 years **Study group 1: 38 (f/u 25) Study group 2: 41 (f/u 27) Control group 1: 39 (f/u 27) Control group 2: 37 (f/u 20) Test** self-report; urinary cotinine levels | **Main results** (all *p*'s > 0.05) • Abstinence at 12 weeks: 47.4% hypnotherapy (1), 43.9% hypnotherapy (2), 28.2% controls (1) • Abstinence at 26 weeks: 36.6% hypnotherapy (1), 34.2% hypnotherapy (2), 18% controls (1) **Other** Non-smokers at 12 and 26 weeks-higher smoking-related self-efficacy at baseline ($p < 0.05$) | - motivation may be influenced by current/recent smoking-related admission - hypnotic suggestions not standardized |

**Table 3.** *Cont.*

| Treatment | Participants and Tests | Results | Limitations |
|---|---|---|---|
| 13 **Shestopal and Bramness, 2019, Norway:** Alcohol Use Disorder/Inpatient (randomized) [61] | | | |
| **Hypnotherapy** (5 × 60 min, wkly) Standard treatment + Erickson's (permissive) hypnosis Relaxation and breathing exercises + mental picture of peaceful places + visualization of mastery over a problem (patient-specific) Hypnotherapist: Erickson's hypnosis training, 10 y experience **Control** (5 × 60 min, weekly) Standard treatment + Motivational interviewing Standard treatment: 5 h group therapy 5 days/week + family therapy session + group activities (trips, walks, movies) **6 weeks total** | *n* = 31 inpatients diagnosed with AUD **Study group: 16 (f/u 13)** **Control group: 15 (f/u 11)** **Test** (baseline, 1 y) MINI psychiatric interview; Alcohol Use Identification Test (AUDIT); Timeline Follow-Back (TLFB) days of abstinence; Global Severity Index (GSI) for mental distress (from Hopkins Symptoms Checklist (HSCL-25) | **Main results** (all *p*'s > 0.05) • total abstinence: 82% hypnotherapy; 54% controls • reduction in alcohol units in previous month: hypnotherapy from 335.6 to 11.7; controls from 291.4 to 14.2 • AUDIT: −88% hypnotherapy, −63% controls **Biopsychosocial outcomes** GSI mental distress ($p > 0.05$): hypnotherapy −0.75, controls −0.46 **Other** intention to treat model: no group differences | - hypnotherapy not a standalone treatment - comprehensive standard treatment program applies to both groups |

Between-group effects were non-significant when comparing HT with psychotherapy for opioid addiction [55], cognitive–behavioral therapy (CBT) or stress management treatment for AUD patients [60,61] and diverse interventions for smoking cessation, including counselling [56], relaxation [59], psychoeducation [51,57], nicotine replacement [56], the "focused cessation technique" [57], or a single HT introductory lecture [50].

Only three studies reported significant between-group differences, with HT alone producing larger benefits than no treatment for smoking [59]; HT with counselling and supportive phone calls associated with considerably higher smoking abstinence rates than supportive phone calls alone [54]; and HT with psychotherapy linked to considerably higher rates of successful methadone withdrawal, lower illicit drug use, and lower withdrawal symptoms than psychotherapy alone [58].

Two studies considered the roles of practice regularity and hypnotic susceptibility in their analysis. Regular HT practice led to significant improvements in self-esteem, anger, and impulsivity in AUD compared with CBT therapy or stress management, which had significantly better outcomes on these variables than irregular HT practice [60]. In another study, when analyzing only participants with high hypnotic susceptibility, differences between HT and relaxation in smoking reduction became significant [59]. Similarly, high hypnotic susceptibility was a predictor of better substance use outcomes [50] and biopsychosocial outcomes [60] wherein participants reported high hypnotizability scores, but not in a study with generally low hypnotizability scores [52].

### 3.3.2. HT Limitations

HT was predominantly used as an adjunct therapy rather than a stand-alone treatment, e.g., in [55,56,58]. The majority of reviewed studies did not use standardized, reproducible hypnotic induction [49,51,54–56,60], while others lacked any clear description of the procedure [53,57–59]. Whether achieving a hypnotic state is linked to clinical outcomes is unclear based on the evidence from the four studies that addressed it [50,52,59,60] and is

inconclusive across the literature more generally [62,63]. On the other hand, relaxation, which is commonly an integral component of HT, has been shown to play an important role in alleviating addiction symptoms, e.g., in [64], and some studies have found it to be the most useful component of HT intervention, e.g., [49].

Several studies used physiological markers of substance use without validated measures of addiction [55–58], and several others lacked both [49,50,53,59]. Potential participant bias was identified in studies that recruited only easily hypnotizable participants [49] or employees during smoking ban [53], or required a participation fee for HT intervention [49,50]. In several studies, group selection was voluntary and not randomized [50,60], or no control was employed [49,52,53].

## 4. Discussion

### 4.1. Substance Misuse Treatment Efficacy

Based on the studies reviewed above, PP and TM appear to be promising treatments for substance abuse and are linked to improved psychosocial well-being in patients receiving treatment for addiction; in contrast, there is little support for the efficacy of HT in treating addiction.

Reviewed studies demonstrated predominantly significant and moderate, e.g., in [27], or substantial, e.g., in [20], efficacy of PP across various classical psychedelics and related pharmacotherapies (LSD, psilocybin, ketamine, ibogaine) in the treatment of a range of substance dependencies (cocaine, opiate, alcohol, tobacco) when compared with psychotherapeutic controls, low-dose active comparisons, and benzodiazepine. Across six open-label studies, significant moderate improvements in abstinence rates were found from baseline to follow-up (ranging from 6 to 12 months). In one qualitative study, moderate reductions in substance use and craving were reported. Studies demonstrating a reduction in addictive behavior also found significant, predominantly moderate improvements across various biopsychosocial outcomes, ranging from comorbid psychiatric diagnoses to regulation of the sense of self and internalization of locus of control and improved relationships.

Similarly, TM led to significant improvements from baseline (6 of the 10 studies) across addiction outcomes for the use of tobacco, alcohol, marijuana, opiates, or stimulants (two studies' results were not significant, and significance was not reported in another two studies). These effects were either moderate or large. TM performed better than a range of control interventions, including various psychotherapeutic approaches and psychotropic treatments, and between-group differences were significant in 7 of the 10 studies. This was true for all reviewed cohorts (veterans, youth, inpatients, outpatients). Moreover, TM reliably improved broader biopsychosocial factors and alleviated psychological symptoms that are recognized risk factors for addictive behavior. When measured, biopsychosocial outcomes were significantly improved from baseline and when compared with control conditions in all but one study.

In contrast, evidence for the efficacy of HT in treating addictions was weak, showing no advantage over comparison addiction interventions and only small advantages over no intervention. Disparities in study design, substantial shortage of information, and questionable motivations of participants were key limitations in assessing the efficacy of HT.

Whilst HT and TM studies reported no adverse events or side effects, PP can be associated with adverse events (usually minor) and requires considerable training of the associated clinicians and care before, during and after the therapy. While the classical psychedelics are physiologically very safe, one fatal event occurred as a direct result of administration of Ibogaine, although post-mortem examination could not identify a clear pathology that was the cause of death [28].

### 4.2. Biopsychosocial Benefits

Both PP and TM led to improvements across a wide range of emotional, cognitive, interpersonal, and intrapersonal factors. Conversely, biopsychosocial outcomes are rarely

considered in HT interventions, highlighting a difference in approach to TM and PP studies. In line with the current understanding of the psychosocial determinants of addiction, improvements in addictive behaviors following PP and TM may be related to the psychosocial benefits associated with those two interventions.

Indeed, the non-directive approach of PP and TM may afford a range of psychosocial benefits through supporting a patient-led process, and an emphasis on the individual's particular acute and post-acute experiences and insights. In contrast, HT induces an ASC in order to provide an intervention that explicitly addresses the target behavior. That is, HT for addiction is typically constrained to a specific problematic behavior, while PP and TM emphasize useful and meaningful ASCs that may produce a wide range of psychosocial changes. This latter emphasis may allow for TM and PP to address more fundamental factors in addictive behaviors than HT, and thereby lead to better results.

### 4.3. Study Quality and Limitations

Early PP studies lack sufficient methodological rigor, with inconsistent dosing regimens, lack of clarity on outcome measures and poorly reported statistical analysis. In contrast, the more recent, well-controlled and analytically rigorous studies used small and heavily screened samples (for some good reasons), limiting generalizability to a broader patient population. Moreover, placebo blinding is difficult across most high-dose PP trials, and patient self-recruitment alongside substantial positive public interest in PP all potentially contribute to inflated expectancy effects.

Various concerns around corporate and quasi-religious motivations have been associated with TM [65]. Tight corporate control of certification, aggressive marketing, overly positive reports, substantial financial gains for the teachers, and promotion of supernatural abilities (e.g., levitation) have contributed to strong critique of the TM community. While results reviewed here are promising, most of the TM studies (7 of 10) suffered from potential conflicts of interest. Moreover, similarly to PP, older TM studies showed less methodological rigor, often using unvalidated tools to measure intervention outcomes, and failing to provide a clear, reproducible account of the full procedure, for example, in relation to control interventions.

Finally, HT interventions for substance addiction showed weak and mixed findings. Inconsistencies in induction procedures and definitions of hypnotic state limit conclusions that can be drawn from this research. Of note, the commonly hypothesized link between hypnotic depth and therapeutic effects has not been established, e.g., in [52,62,63]. Also, participant experiences often appear indistinguishable from those undergoing relaxation interventions [66], and when identical interventions are labelled as either relaxation or hypnosis, reports of suggestibility and involuntariness increase more in the latter [67], suggesting that outcomes may in part depend on expectancy effects or demand characteristics. These limitations suggest that where substance misuse benefits have been reported in a minority of studies, these may be accounted for by factors like suggestibility or therapeutic alliance, rather than the hypnotic state.

Lastly, in contrast to many modern PP studies, both TM and HT are unreliable in their ability to induce an ASC, and the majority of reviewed studies failed to include markers of achieving an ASC. Therefore, it is difficult to assess whether the production or characteristics of an ASC during these interventions was associated with beneficial outcomes.

### 4.4. Future Directions

Many of the studies reviewed herein failed to measure acute altered state experience, making it difficult to evaluate whether and which of these may be related to beneficial addiction treatment outcomes. However, contemporary PP literature has found that features of the acute subjective psychedelic experience, such as oceanic boundlessness, ego dissolution or universal interconnectedness, most often measured by Mystical Experiences Questionnaire (MEQ) or Five Dimensions of Altered States of Consciousness Scale

(5D-ASC) are one of the most robust predictors of clinical benefit across a number of indications including cancer-related distress [68], depression [69], or substance abuse [22]. These and similar measures should be incorporated across interventions such as TM and HT to determine whether certain kinds of altered state experiences are important for producing clinical benefit. Well-validated ASC measures and constructs should be used to assess a wide range of features commonly associated with ASC experience, thereby increasing specificity and consistency across studies.

Indeed, mental health benefits appear to be linked to subjective features of the acute ASC experience across a number of ASC interventions [70]. Therefore, it may be useful to consider the utility for various other ASC induction methods in the treatment of addictions, including sensory deprivation, neurofeedback, tactile sound systems, hypnagogic lights, virtual reality, and so on. If beneficial, they could offer a feasible adjunct to current treatments. Moreover, potential synergistic effects of combining two or more ASC interventions should be explored [71].

While some studies have found better clinical outcomes following a longer period of meditation, e.g., [72], or a higher number of psychedelic doses [27], typically, the duration of meditation has been short, e.g., only two of the reviewed TM studies provided follow-up sessions for longer than 6 months [43,46] and the number of psychedelic doses was limited, e.g., one to three doses, [27]. Future studies could extend program duration or number of sessions to determine optimal program length using designs that can assess cumulative effects. Adding an assessment of cost effectiveness would further contextualize optimal program length within economic considerations.

Lastly, it has been speculated that an inherent need to achieve altered states may motivate drug-seeking behaviors that can lead to addiction [10,73]. Future work should explore this claim, and then determine whether providing beneficial ASC interventions (including those induced by psychedelic drugs) could mitigate the need to engage in harmful drug use that can lead to dependence and addiction. One implication of this could be to expand the use of ASC interventions from treatment to prevention.

## 5. Conclusions

The effects of PP on substance misuse outcomes indicated promise and warrant further, more rigorous investigation. Results of TM were also encouraging, and findings should be replicated by researchers with no ties to TM institutions. In contrast, and despite some beneficial effects, evidence suggested that HT may not be an effective treatment for substance misuse. Moreover, PP and TM reliably led to biopsychosocial improvements (e.g., reductions in stress, anxiety, or depression; improved coping skills and interpersonal relationships; increased self-efficacy or sense of purpose) that were not observed in comparison conditions in this review. Some of those benefits, including renewed sense of self or increased understanding of the meaning of life, are not commonly reported in other trials of addiction treatments [74]. A key question is whether certain subjective aspects of the acute ASC associated with both PP and TM are important for driving clinical change. Moreover, given the potential role of biopsychosocial change in successful treatment of addiction [75], future research could investigate whether the impact of ASC interventions on substance misuse is mediated by improvements in biopsychosocial functioning.

**Author Contributions:** Conceptualization, A.D.S.; methodology, A.D.S. and P.P.; data curation, A.D.S. and P.P.; writing—original draft preparation, A.D.S. and P.P.; writing—review and editing, P.L.; supervision, L.D. All authors have read and agreed to the published version of the manuscript.

**Funding:** This research received no external funding.

**Data Availability Statement:** No new data were created or analyzed in this study. Data sharing is not applicable to this article.

**Conflicts of Interest:** Liknaitzky has received research funding from Incannex Healthcare Ltd, the Multidisciplinary Association for Psychedelic Studies, and Beckley Psytech, is a member of the Medical Advisory Board of Incannex Healthcare Ltd, and the Scientific Advisory Board of The MIND

Foundation. These organizations were not involved in any aspect of this paper, including the decision to write it, drafting the paper, or its publication. PP and AS are co-founders of Enosis Therapeutics, a startup developing virtual-reality based mental health programs for diverse psychotherapeutic approach, including psychedelic-assisted psychotherapy and other altered-state based therapies. LD declares no conflicts of interest.

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
