# Peer review of "Producing Altered States of Consciousness, Reducing Substance Misuse: A Review of Psychedelic-Assisted Psychotherapy, Transcendental Meditation and Hypnotherapy"

_psychoactives, doi:10.3390/psychoactives3020010_

Round 1
Reviewer 1 Report
Comments and Suggestions for Authors
Even the topic seems interesting, paper is not providing any scientifically relevant idea for future research. Method described in the paper about data they obtained have several flaws and results itself in the relevant papers are contradictory. It is very hard to understand and comprehend for reader due to heavy language about therapy for readers. Even authors clearly explained their limitations lack of supporting data about psychedelics make paper out of scope for the journal.
Author Response
We would like to know what the flaws in the methods used to obtain the data are, to make it possible for us to address them.
The fact that results are contradictory is indeed one of the key reasons for writing the review and one of the key findings/ limitations that are outlined in the paper.
We leave it to the editorial team to assess whether the complexity of the language used is beyond the grasp of or in keeping with the linguistic standards that this journal expects of the papers it publishes. We do point out that the first author's mother tongue is not English and she wrote this with English being her second language.
We do find that evidence for psychedelics is greater than for the other two methods. We also believe that comparing psychedelics with other modalities that claim a similar psychological mechanism is of importance to the field which is growing very fast and has limited viable control methods.
Reviewer 2 Report
Comments and Suggestions for Authors
This is an excellent review. The only change I recommend is to provide further detail on the assertion on p. 26 lines 449-451 of how certain features of the psychedelic experience are associated with clinical benefits. I would like to see those features specified with links to the relevant studies. (I suspect that a mystical experience is one of these features.)
Author Response
Thank you. The manuscript has been revised as suggested and the suggested section now reads:
'Many of the studies reviewed herein failed to measure acute altered state experience, making it difficult to evaluate whether and which of these may be related to beneficial addiction treatment outcomes. However, contemporary PP literature has found that features of the acute subjective psychedelic experience, such as oceanic boundlessness, ego dissolution or universal interconnectedness, most often measured by Mystical Experiences Questionnaire (MEQ) or 5 Dimensions of Altered States of Consciousness Scale (5D-ASC) are one of the most robust predictors of clinical benefit across a number of indications including cancer-related distress (Ross et al., 2016), depression (Roseman et al., 2018), or substance abuse (Bogenschutz et al., 2015).'
Reviewer 3 Report
Comments and Suggestions for Authors
The paper is a systematic review of the literature on the use of psychoactive substances in psychotherapy, meditation, and hypnotherapy. The authors have followed the PRISMA guidelines and reported findings that might be helpful for future studies. However, I think there are some aspects that require clarification:
- it is not clear to me if the aim of the authors was to evaluate the misuse of psychoactive substances in PP, TM, and HT, or their therapeutic add-ons. I think the introduction should be revised (lines 54-59)
- systematic review guidelines suggest including an evaluation of the quality of the paper included. This might be done by authors even if they already discussed this aspect in their discussion.
- your discussion is structured mainly on the separation between techniques. Have you evaluated between techniques the effects on specific goals?
Author Response
Thank you, below is the response to the comments:
1. The introduction has been revised as suggested to clarify the goals of this review and now reads:
'A wide range of ASC induction methods have been used to treat different forms of substance use disorders within ceremonial, self-medicating, and clinical settings (Vaitl et al., 2005). This review explores three ASC therapies that have been empirically explored as treatments for substance misuse to a greater degree than other interventions: psychedelic-assisted psychotherapy (PP), Transcendental Meditation (TM), and hypnotherapy (HT).'
2. The three methods that have been looked at in this review have since a shift in application over the years, with studies spanning 50 years and consequently a big change in scientific quality. That, as well as the highly diverse study designs within each method, and the differences between methods themselves, makes it very difficult to apply the same quality measurement tool across all reviewed papers. Instead, the authors thoroughly described all limitations and risks of bias for each modality.
3. Same as above, the goals of studies have been so varied that identifying goals that have been common in both the substance type and assessment would be artificial and commonalities would have to be stretched. In fact, initially the authors began to write this review with the goal of comparing specific psycho-social outcomes of all modalities, however the evidence that was eventually included after all exclusion criteria were applied made for insufficient evidence for those outcomes.
Round 2
Reviewer 1 Report
Comments and Suggestions for Authors
I think in that format it is not appropriate and in the scope of journal.
Author Response
Increased interest in altered states treatments calls for a need to assess and compare definitions, approaches and taxonomy of methods that proclaim the induction of an Altered State of Consciousness (ASC). This paper was driven by the need to compare treatments that use very different methods and protocols, yet are nonetheless classified through the unifying lens of an ‘ASC’ treatment.
Given that we have reviewed the evidence from three different treatment modalities in order to compare them, we understand that the format of presented outcomes may be less familiar than if only one modality was reviewed.
However, authors of this review followed the PRISMA guidelines and the paper contains evidence for psychedelic-based treatments as one of the three key approaches explored, therefore we believe that it is the right format for the journal.